# A Lightweight RFID Mutual Authentication Protocol with PUF

**DOI:** 10.3390/s19132957

**Published:** 2019-07-04

**Authors:** Feng Zhu, Peng Li, He Xu, Ruchuan Wang

**Affiliations:** 1School of Computer Science, Nanjing University of Posts and Telecommunications, Nanjing 210023, China; 2Jiangsu High Technology Research Key Laboratory for Wireless Sensor Networks, Nanjing 210003, China

**Keywords:** lightweight, radio frequency identification, authentication, physical unclonable function, security, privacy

## Abstract

Radio frequency identification is one of the key techniques for Internet of Things, which has been widely adopted in many applications for identification. However, there exist various security and privacy issues in radio frequency identification (RFID) systems. Particularly, one of the most serious threats is to clone tags for the goal of counterfeiting goods, which causes great loss and danger to customers. To solve these issues, lots of authentication protocols are proposed based on physical unclonable functions that can ensure an anti-counterfeiting feature. However, most of the existing schemes require secret parameters to be stored in tags, which are vulnerable to physical attacks that can further lead to the breach of forward secrecy. Furthermore, as far as we know, none of the existing schemes are able to solve the security and privacy problems with good scalability. Since many existing schemes rely on exhaustive searches of the backend server to validate a tag and they are not scalable for applications with a large scale database. Hence, in this paper, we propose a lightweight RFID mutual authentication protocol with physically unclonable functions (PUFs). The performance analysis shows that our proposed scheme can ensure security and privacy efficiently in a scalable way.

## 1. Introduction

Radio frequency identification technology, which can obtain the information and identify objects automatically through wireless channel, is a fundamental technology for the Internet of Things (IoT) [1]. Since RFID is able to work well even in many severe environments without any artificial interference, it has been widely used in many fields such as asset tracking, supply chain management, logistic control, and so on [2]. A typical RFID system consists of tags, readers, and a backend database, as shown in Figure 1.

In general, each tag has its own unique identity (ID) and is attached to an object. In order to recognize an object, the reader needs to launch an identification procedure to identify the tag ID though the wireless channel between the reader and the tag. After the reader obtains the tag ID, it can further retrieve the related data of the object from the backend database with the tag ID. However, due to the weakness of wireless communication, RFID also suffers from security issues.

There are two main international RFID standards bodies: International Standards Organisation (ISO) and Electronics Product Code Global Incorporated (EPCglobal). EPCglobal designed EPC Class-1 Generation-2 standard (short for EPC Gen2) for RFID products [3], which was approved as ISO18000-6C in 2006. EPC Gen2 tags are the mainstream for RFID applications [4]. Specifically, for the purpose of large-scale and extensive range applications, passive tags are the most widely used kind of EPC Gen2 tags, which usually have low costs and thus have limited computation power and storage size. Therefore, such tags only support some restricted operations such as exclusive OR (XOR), random number generator (RNG), and so on.

In recent years, there are many kinds of attacks aiming at RFID system (i.e., tag tracing, impersonation, desynchronization attacks, physical attacks [5], and clone attacks). The security and privacy of RFID systems are threatened by these attacks. Because of the hardware limitation of an EPC Gen2 tag, the security problem becomes even more complicated. There are no more than 2000 logical gates available for the security in a passive tag. However, the traditional security mechanisms like cryptographic functions need too much hardware resources to support. For example, the simplest implementation of Advanced Encryption Standard (AES) algorithm requires more than 3400 logical gates [6], while the traditional hash functions such as Secure Hash Algorithm 256 (SHA-256), Message-Digest Algorithm 4 (MD4), and Message-Digest Algorithm 5 (MD5) need 7350–10,868 logical gates to support [7]. Fortunately, several lightweight hash functions, such as PHOTON [8] and SPONGENT [9], have been introduced, which are able to provide security primitives for RFID tags with low hardware cost. For instance, a 128-bit SPONGENT hash function can be implemented with 1060 logical gates [9]. Researchers have proposed many protocols for authenticating low-cost RFID tags in RFID systems. However, they have certain limitations, weaknesses and vulnerabilities. For example, HB-family protocols [10,11,12] are based on matrix multiplication and some XOR operations. MAP-family protocols [13,14,15] rely on bitwise operations like AND, XOR and OR. They have good performance in terms of computation, cost, and overhead. However, these protocols suffer from various security vulnerabilities such as desynchronization attacks, physical attacks, and counterfeit [16,17,18,19,20]. Later, an ultra-lightweight authentication protocol called Strong Authentication and Strong Integrity Protocol (SASI) [21] was proposed to provide strong authentication and strong integrity, while researchers found that it is vulnerable to tag tracing and desynchronization attacks [22,23,24]. To address the drawbacks of SASI protocol, Gossamer protocol was introduced in Peris-Lopez et al. [25]. Unfortunately, this protocol is still vulnerable to desynchronization attacks [26]. Recently, an ultra-lightweight authentication protocol for mobile commerce [27] was proposed, employing only simple operations such as XOR, addition modulo, and shift operation. However, Aghili, and Mala [28] revealed that reader impersonation and physical attacks can break this scheme. In addition, in the above protocols tags are threatened by clone attacks, thus calling for a proper solution.

In this work, we propose novel lightweight RFID mutual authentication protocols based on physically unclonable functions (PUFs), which is able to meet the security demands of RFID system and satisfy the hardware limitation of low-cost tags. The rest of the paper is organized as follows. Section 2 introduces the PUF technique. Section 3 discusses some related work in PUF-based RFID protocols. Section 4 presents our new lightweight protocols. Section 5 performs the security analysis of the proposed protocols. Section 6 analyzes the protocol performance. Section 7 discusses the implementation of the proposed protocols. Section 8 summarizes the paper.

## 2. PUF Technique

PUF is a novel technique for the semiconductor security, which serves as a unique identity for a semiconductor device such as chips. In 2002, the first PUF, based on optics, was invented [29]. Then the PUF and the measurement circuit are integrated together on the same electrical circuit [30]. After that, PUFs have been adopted in security authentication [31,32] and intellectual property protection [33]. With the rapid development of IoT, PUF has a good perspective in the safeguarding and anti-counterfeiting area.

PUF relies on physical microstructure that occurs naturally during chip manufacturing [34]. The physical microstructure depends on random physical factors that are unpredictable and uncontrollable. Therefore, it is impossible to produce two chips that are exactly same, which indicates that PUF techniques have good resistance against the clone attack [35]. One of the most classic PUF architecture is arbiter PUF [36]. As demonstrated in Figure 2, the arbiter PUF circuit consists a signal transmission delay circuit and an arbiter. The circuit has 64 switches. Each switch has one control signal terminal, two input terminals, and two output terminals. When the input signal (b_0_ to b_63_) arrives, the circuit will generate two transmission delay path with equal length. The input signal is then sent to the two paths at the same time. The arbiter will finally make a decision based on its two inputs. 

Due to the random variations of physical microstructures, different PUFs will have different outputs with the same challenge. Since these physical microstructures are generally unclonable, the behavior of PUF is hard to predict or extract. Therefore, it is very difficult for attackers to forge a PUF circuit. Furthermore, when comparing with a cryptographic function, a PUF needs less hardware resources, which is suitable for a passive tag. For example, an arbiter PUF circuit requires roughly eight logical gates per input bit plus four logical gates for the arbiter. Since such a primitive will produce just one single output bit, to construct a k-bit response, one lightweight solution is to feed a linear feedback shift register (LFSR), which needs an extra four gates per bit, with the input challenge to generate a pseudorandom sequence. Then the PUF is evaluated k times using k different bit vectors from the pseudorandom sequence [36]. Thus, a 64-bit arbiter PUF (with a LFSR) needs about 784 logical gates, which is much less than the encryption circuits such as MD5, AES, and so on. With the rapid growth of IoT, there is an urgent need of security authentication and confidential data protection. RFID tags integrated with PUF, which utilizes both the unforgeable property of PUF and RFID authentication, is a choice with great potentiality to improve the security and privacy in IoT.

## 3. Related Work

With the wide application of RFID technology, many researchers try to leverage PUF technique to improve the security of RFID system. Early works like [37,38,39] most rely on the distinct challenge–response pairs (CRPs) generated by PUF to authenticate tags. When verified by the server, only the proper tag can provide the correct response. Unfortunately, they face the problems of tag tracing, lack of forward secrecy, and desynchronization attacks. 

In regard of ensuring the untraceability property, most of RFID authentication protocols trade-off performance for the sake of privacy. The most common method is to encrypt the tag identity with a cryptographic function. In this case, the reader (or the backend server) is required to perform an exhaustive search to verify the tag, which may result in a poor performance. For example, Akgun and Caglayan [40] designed a PUF-based authentication protocol, in which the tag identity is encoded by a hash function. To verify a tag, upon receiving the hash code, the backend server has to perform a linear search of all tags in the system, computing each possible hash code with the transmitted nonce until a match is found. Worse than that, the scheme is lack of forward secrecy. A similar protocol was proposed by Aysu et al. [41], in which the encoding is done with a pseudo random function. Besides, their protocol can support forward secrecy. However, Huth et al. [42] revealed that this protocol can leak sensitive data potentially and is vulnerable to tracing attacks. They made up this flaw in their enhanced protocol but an additional channel-based key agreement phase is needed. Furthermore, their protocol still relies on the exhaustive search for validation. Hence, these schemes are not scalable for applications with a great number of tag identities in the database.

According to Burmester et al. [43], if the backend server is able to find the tag identity in the database by the received data directly, which means that no computation (i.e., the hash function or the cryptographic function) is needed before checking each record, the lookup time cost can be constant. Thus, some protocols [44,45,46,47] leverage temporary identities instead of the original tag identity during authentication to avoid tag tracing. The protocol [44] integrates PUF and linear feedback shift register together, which achieves the goal of lightweight authentication and is suitable for low-cost RFID tags. In this protocol, PUF is used as a mask-generator to hide messages between the reader and the tag. However, Xu et al. [45] revealed that it has security flaws of data confidentiality and desynchronization attacks. They proposed an improved protocol with PUF and simple operations (i.e., loop left shifting, AND, and XOR) only, which is more efficient than the protocol [44]. Unfortunately, Bendavid et al. [46] figured out that a sophisticated attacker can still break the scheme [45] by a desynchronization attack or a secret disclosure attack. Their enhanced scheme introduces a hash function and can avoid such attacks. However, since the tag needs to store secret parameters in the memory, this protocol cannot guarantee security against physical attacks. Not only Bendavid et al. [46] but most of the existing schemes also keep secret parameters in tag memory. For instance, all the schemes mentioned above do so except Aysu et al. [41] and Huth et al. [42]. Similar to them, Gope et al. [47] stores no secret parameters in tag memory but only a temporary identity and a set of unique unlinkable pseudo-identities, which are also shared by the backend server to prevent desynchronization attacks. However, since each desynchronization will consume one pseudo-identity, when the pseudo-identities are depleted, the tag has to execute the setup phase with the backend server through a secure channel to obtain the new set of pseudo-identities and temporary identity. 

This paper aims to address the issues of performance and security in previous works. Since PUFs can be classified into two categories, ideal PUFs and noisy PUFs, based on whether the PUF response is stable, we first propose a lightweight RFID mutual authentication protocol based on ideal PUF. Then we enhance this protocol so that it can be adopted with a noisy PUF. We summarize our contributions as follows:

Firstly, our proposed protocols can meet several crucial security requirements such as untraceability, mutual authentication, resilience of desynchronization, forward secrecy, and unclonability, which are all important in RFID systems. 

Secondly, our proposed protocols can ensure the security requirements efficiently with good scalability. Specially, in our protocols, the tag does not store any secret parameter, which not only reduces the storage cost, but also enhances the security. Moreover, there is no need for the backend server to perform exhaustive search. In addition, our protocols also support low-cost RFID tags.

## 4. Proposed Protocol

In this section, we present our proposed RFID mutual authentication protocols. Before describing our protocols, we briefly discuss the adversary model and underlying assumptions.

### 4.1. Adversary Model

We consider the adversary as a Dolev-Yao intruder [48], who eavesdrops the insecure communication channel between a reader and a tag. The adversary can alter messages and also be able to block messages from a reader to a tag or vice versa. In addition, the adversary has the capability to launch any physical or clone attacks.

### 4.2. Assumptions

The assumptions for our proposed protocols are presented below.

First, each RFID tag is embedded with a PUF, where the PUF output depends on its unique physical characteristics. If an attacker tries to tamper with the PUF, the behavior of the tag will change, which makes the tag useless. 

Second, each RFID tag contains a hash function and so does the backend server. Given the same input, the hash function at server side should generate the same result as the one on a tag.

Third, there is a secure channel, inaccessible by the adversary, between the reader and the backend server. As illustrated in Figure 3, in a typical RFID system, the communications between the reader and the tag are using a public wireless channel, and are thus considered to be insecure. On the contrary, the reader is connected the backend server through a wired connection. Not only that, the communications between the backend server and the reader are generally on a private intranet, which are considered to be secure [49]. Therefore, our proposed protocols just need to focus on the insecure communications between the reader and the tag. 

Fourth, RFID tags have limited resources of computation power and storage, while backend servers are trusted and have no such limitation. 

### 4.3. Notations

The operations and cryptographic functions used in our proposed protocols are summarized in Table 1 for the convenience of the understanding. 

### 4.4. Proposed Ideal PUF-Based Authentication Protocol 

In this section, we propose an authentication protocol based on ideal PUF, which requires that the PUF response should be stable. An ideal PUF can be considered as a challenge–response pair. Thus, in tag T with an ideal PUF, the response R of a challenge C is donated as R = PUF^T^(C). Our proposed protocol consists of two phases: setup phase and authentication phase, which are demonstrated as follows.

#### 4.4.1. Setup Phase

When a new tag T joins the RFID system, the server and the tag need to negotiate with each other as follows. For each tag, the setup phase needs to be executed only once through a secure channel. The entire setup phase is demonstrated in Figure 4.

**Step 1:** S->T: {C_1_}.

The server generates a challenge C_i_ and then sends C_1_ to tag T. 

**Step 2:** T->S: {R_1_}.

After receiving the challenge C_1_ from S, tag T produces the response R_1_ by using its physically unclonable function PUF^T^ and sends R_1_ to S. 

**Step 3:** S->T: {SID_1_^T^}.

Upon receiving the response from T, S generates a unique session identity SID_1_^T^ for the first round authentication and sends it to T. At last, the server stores {SID_1_^T^, C_1_, R_1_} for tag T while tag T stores the session identity SID_1_^T^. 

#### 4.4.2. Authentication Phase

Since each reader is connected to the backend server through a secure connection, the reader and the server can be considered as a single unit S. Thus, we omit the reader in the authentication. Figure 5 shows the i-th round mutual authentication among the tag, reader, and server. The detailed steps are presented below.

**Step 1:** S->T: {Hello}.

The server sends a “Hello” message to tag T to initiate the authentication phase.

**Step 2:** T->S: {SID_i_^T^}.

After receiving the authentication request, T transmits the session identity SID_i_^T^ to S.

**Step 3***:* S->T: {C_i_, N’, Auth_1_}.

Upon receiving the session identity SID_i_^T^ from T, S reads C_i_, R_i_ from the corresponding entry for SID_i_^T^. After that, S generates a random number N and computes N’ and Auth_1_, where N’ = R_i_ ⊕ N, Auth_1_ = Hash(N||R_i_) and sends {C_i_, N’, Auth_1_} to T. If SID_i_^T^ is not in the database, the authentication phrase will be terminated since T may be an invalid tag. 

**Step 4***:* T->S: {R’, M’, Auth_2_}.

Once receiving the response from S, tag T supplies the challenge C_i_ to its PUF to produce a response R_i_ and computes N = N’⊕R_i_ in order to verify the server authentication parameter Auth_1_. If the server is verified, T first generates a random number M and then computes C_i+1_ = Hash(N||M||C_i_), R_i+1_ = PUF^T^(C_i+1_), R’ = R_i+1_⊕N, M’ = M⊕N, Auth_2_ = Hash(N||M+1||R_i+1_) and composes a reply {R’, M’, Auth_2_} to S. 

**Step 5:** S->T: {Auth_3_}.

After receiving the response from T, S computes R_i+1_ = R’⊕N, M = M’⊕N and validates the tag authentication parameter Auth_2_. If the tag is a valid one, S computes C_i+1_ = Hash(N||M||C_i_), SID_i+1_^T^ = Hash(N||M+2||SID_i_^T^), Auth_3_ = Hash(N||M+2), and sends {Auth_3_} to T. Then S stores {SID_i+1_^T^, C_i+1_, R_i+1_} in its database from future authentication while {SID_i_^T^, C_i_, R_i_} is still kept in case of desynchronization attacks.

**Step 6:** Validation at Tag T.

Once receiving the response from S, tag T verifies the response parameter Auth_3_. If correct, T computes SID_i+1_^T^ = Hash(N||M+2||SID_i_^T^) and updates the current session identity SID_i_^T^ to SID_i+1_^T^ for the next round mutual authentication.

If there is any validation failure among previous steps, it means that either S or T may be compromised. Thus, the authentication phase should be terminated. Otherwise, the successful mutual authentication indicates that both S and T are valid and they can continue communicate with each other.

### 4.5. Proposed Noisy PUF-Based Authentication Protocol 

In Section 4.4, we made an assumption that our proposed protocol is used in the ideal PUF environments. However, although most of PUF-based protocols rely on the same assumption, lots of existing PUFs are not that perfect. Thus, we need to enhance our proposed protocol so that it can work with a noisy PUF. Recently, a few noisy PUF-based schemes such as Aysu et al. [41] and Huth et al. [42] are proposed in recent years. Similar to them, we also utilize the fuzzy extractor in our enhanced protocol. A fuzzy extractor can be denoted as FE: = (FE.Gen, FE.Rec). The FE.Gen algorithm takes a variable z as input and outputs randomness r and helper data hd. The FE.Rec algorithm recovers r from an input z’ and the helper data hd if the hamming distance between the z and the z’ is sufficiently small.

#### 4.5.1. Setup Phase

Similar to our ideal PUF-based protocol, when a new tag T joins in the RFID system, the tag needs to be registered into the backend server via a secure channel. The setup phase of the enhanced protocol is shown in Figure 6.

**Step 1:** S->T: {C_1_}.

The server generates a challenge C_i_ and then sends C_1_ to tag T. 

**Step 2:** T->S: {R_1_}.

After the challenge C_1_ is received, tag T produces the response R_1_ by its PUF^T^ and sends R_1_ to S subsequently. 

**Step 3:** S->T: {SID_1_^T^}.

Once receiving the response R_1_, S generates a unique session identity SID_1_^T^ for the first round authentication, computes (r_1_, hd_1_) = FE.Gen(R_1_), and then sends {SID_1_^T^, hd_1_} to T. At last, the server stores {SID_1_^T^, C_1_, r_1_} for tag T, while tag T stores the session identity and the helper data. 

#### 4.5.2. Authentication Phase

Similar to our proposed ideal PUF-based protocol, we consider the reader and the server as a single unit S. The authentication phase of the enhanced protocol is presented in Figure 7, consisting of the following steps.

**Step 1:** S->T: {Hello}.

The server sends a “Hello” message to tag T to initiate the authentication phase.

**Step 2:** T->S: {SID_i_^T^}.

Upon receiving the authentication request, T reads its memory and sends the session identity SID_i_^T^ to S.

**Step 3:** S->T: {C_i_, N’, Auth_1_}.

After receiving SID_i_^T^ from T, S reads C_i_, r_i_ from the matched entry. Then S generates a random number N and computes N’ and Auth_1_, where N’ = r_i_⊕N, Auth_1_ = Hash(N||r_i_) and sends {C_i_, N’, Auth_1_} to T.

**Step 4:** T->S: {R’, M’, Auth_2_}.

Once receiving the response from S, tag T supplies the challenge C_i_ to its PUF to produce a response R_i_, computes r_i_ = FE.Rec(R_i_, hd_i_), N = N’⊕r_i_ to verify the server authentication parameter Auth_1_. If the server is verified, T first generates a random number M and then computes C_i+1_ = Hash(N||M||C_i_), R_i+1_ = PUF^T^(C_i+1_), R’ = R_i+1_⊕N, M’ = M⊕N, Auth_2_ = Hash(N||M+1||R_i+1_) and composes a reply {R’, M’, Auth_2_} to S. 

**Step 5:** S->T: {hd’, Auth_3_}.

After receiving the response from T, S computes R_i+1_ = R’⊕N, M = M’⊕N and validates the tag authentication parameter Auth_2_. If the tag is a valid one, S computes (r_i+1_, hd_i+1_) = FE.Gen(R_i+1_), C_i+1_ = Hash(N||M||C_i_), SID_i+1_^T^ = Hash(N||M+2||SID_i_^T^), hd’ = Hash(M||r_i_)⊕hd_i+1_, Auth_3_ = Hash(N||M+2||hd_i+1_) and sends {hd’, Auth_3_} to T. Then S stores {SID_i+1_^T^, C_i+1_, r_i+1_} in its database from future authentication while {SID_i_^T^, C_i_, r_i_} is still kept in case of desynchronization attacks.

**Step 6:** Validation at Tag T.

Once receiving the response from S, tag T first computes hd_i+1_ = Hash(M||r_i_)⊕hd’ and then verifies the response parameter Auth_3_. If correct, T computes SID_i+1_^T^ = Hash(N||M+2||SID_i_^T^) and stores {SID_i+1_^T^, hd_i+1_} instead of {SID_i_^T^, hd_i_} for the next round mutual authentication.

If there is any validation failure among previous steps, the authentication phase should be terminated. Otherwise, the successful mutual authentication indicates that both S and T are valid and they can continue communicate with each other.

## 5. Formal Security Analysis

In this section, we analyze our proposed protocol based on ideal PUF formally on the major security requirements. We can also analyze our enhanced protocol in a similar way.

### 5.1. Adversarial Model

We model the adversary A as a probabilistic polynomial time (PPT) algorithm, who is able to control over the insecure communication channel between a tag T and the reader–server unit S. In other words, the adversary A can access the following oracles only during the authentication phase.Query(T, x_1_, m_2_): A sends x_1_ to T and receives m_2_ from T. This oracle models the adversary’s ability to interrogate tags in the system.Send(S, m_1_, x_2_, m_3_): A sends m_1_ to S and receives x_2_ from S. Then A sends m_3_ to S as a response. This oracle models the adversary’s ability to act as a legitimate tag in the system.Execute(T, S): A eavesdrops on the channel between T and S, and can also tamper with the protocol messages. This oracle models the adversary’s ability to continuously monitor the channel between T and S.Block( ): A blocks a part of the protocol and is able to break the synchronization between T and S. This query models the adversary’s ability to launch a denial of service attack.Reveal(T): A manages to access to the content in the memory of the tag T. The oracle models the adversary’s ability to physically crack a tag and obtain the secrets in its memory.

In the adversarial model, A can call the oracles Query, Send, Execute, and Block any polynomial number of times. However, on the same tag, the Reveal oracle can be called only once. Since once the Reveal oracle is called, the tag is considered compromised. Thus, there is no need to invoke the Reveal oracle more than once on the same tag. 

### 5.2. Security Analysis

Given the above adversarial model, we first present two lemmas states important conclusions that will be used in the following proofs.

**Lemma** **1.**
*The secret parameters of a tag in our ideal PUF-based protocol cannot be exposed even with the Reveal oracle.*


**Proof:** In Step 3, the random number N is delivered to T by transmitting R_i_⊕N, which can be viewed as an encryption of N with the key R_i_. Since R_i_ is unknown to the adversary A, R_i_ acts as a random encryption key so that N will be delivered secretly. Similarly, in Step 4, the secret response parameter R_i+1_ and the random number M are encrypted with the key N and then sent to S. Without knowing N, the adversary A cannot figure out R_i+1_ or M. In the protocol, there is no need for the tag to store any secret security parameters. Thus, even if the adversary A call the Reveal oracle to obtain the secrets from the tag memory, the only data can be obtained is the session identity, which cannot help the adversary A to pass the authentication process. Furthermore, as discussed in Section 4.2, any illegal access to PUF will affect its CRP behavior, which will result in a useless tag.
**Q.E.D.**


**Lemma** **2.**
*The session identities of a tag in the proposed protocol cannot be correlated without calling the Reveal oracle.*


**Proof:** Every (i+1)th round session identity SID_i+1_^T^ is generated from the random numbers N and M, and SID_i_^T^ via a hash function. Since N and M are unknown to the adversary A, the two session identities, SID_i_^T^ and SID_i+1_^T^, cannot be correlated with a non-negligible probability unless the adversary A calls the Reveal oracle.
**Q.E.D.**


#### 5.2.1. Privacy

**Theorem** **1.**
*In the proposed protocol, tags are universally untraceable.*


**Proof:** We model this by a game between a challenger C and the adversary A. Assume C has chosen two tags, T_0_ and T_1_, and a valid reader–server unit S. A starts the game by invoking the Query, Send, Execute, and Block oracles on T_0_, T_1_, and S for a polynomial number of times. A records the outputs of the oracles calls and notifies C. Then C randomly chooses a bit b and sets T = T_b_. Now A calls the oracles Query, Send, Execute, and Block on T and S for a polynomial number of times. After finishing calling the oracles, A outputs a bit b’ and wins the game if b’ = b. The advantages of successfully identifying is defined as Adv_A_ =∣Pr(b’ = b) – 1/2∣. Tags are claimed to be universally untraceable if Adv_A_ is negligible. In the above game, since A does not know the secret response parameters of the i-th round authentication i.e., R_i_. Thus, by Lemma 1, A cannot generate the updated session identity SID_i+1_^T^ for the i+1th round authentication. Furthermore, by Lemma 2, A cannot correlate the session identities to each other. Hence, A can only randomly guess a bit b’ and the probability that Pr(b’ = b) will be greater than 1/2 is negligible. Therefore, the advantage of winning the game, Adv_A_, is negligible. 
**Q.E.D.**


**Theorem** **2.**
*The proposed protocol ensures forward secrecy.*


**Proof:** We model this by a game similar to the universally untraceable one, except that the adversary A will call the Reveal oracle in the last step. Assume a challenger C has chosen two tags, T_0_ and T_1_, and a valid reader–server unit S for the game. A starts the game by invoking the Query, Send, Execute, and Block oracles on T_0_, T_1_, and S for a polynomial number of times. A records the outputs of the oracles calls and notifies C. Hereafter, C carries out the authentication between each tag—T_0_ and T_1_—and the reader–server unit S successfully. Then C randomly chooses a bit b and sets T = T_b_ and gives it to A. Now, A calls the Reveal oracle to obtain all the secrets of T. Finally, A outputs a bit b’ and wins the game if b’ = b. The advantages of successfully identifying is defined as Adv_A_ =∣Pr(b’ = b) – 1/2∣. The proposed protocol is claimed to ensure forward secrecy if Adv_A_ is negligible.By Lemma 1, no secrets will be exposed to A before the Reveal oracle. In addition, since the updated session identity is a Hash function of the outdated one, A cannot inverse the Hash function. Besides, each CRP is randomly generated so that they should be independent to each other. Furthermore, in the proposed protocol, only the session identity is stored in the tag memory. In other word, all the secrets needed in the authentication phase are one-time. Even with the Reveal oracle, the only information A manages to obtain is the session identity which, by Lemma 2, cannot be correlated to the outdated one. Thus, the adversary A has no advantage over random guess and the probability that Pr(b’ = b) will be greater than 1/2 is negligible. So the advantage of winning the game, Adv_A_, is negligible. 
**Q.E.D.**


#### 5.2.2. Mutual Authentication

**Theorem** **3.**
*The proposed protocol accomplishes secure mutual authentication.*


**Proof:** We model this by a game between a challenger C and the adversary A. Assume C has chosen a tag T and a valid reader–server unit S for the game. A starts the game by invoking the Query, Send, Execute, and Block oracles on T and S for a polynomial number of times. A records the outputs of the oracles calls and notifies C. On one hand, assume A attempts to impersonate a tag by the Send oracle. A wins the game if A can be authenticated as a legitimate tag. In the authentication process, A needs to reply a valid session identity SID_i_^T^ and a valid authentication parameter Auth_2_ = Hash(N||R_i+1_) to the reader–server unit S. In order to pretend to be a legitimate tag, A has to know the secret parameters of the i-th round authentication. However, by Lemma 1, N and R_i_ are unknown to A. Therefore, the probability for A to impersonate a legitimate tag is negligible. On the other hand, assume A attempts to impersonate a reader–server unit by the *Query* oracle. A wins the game if A can be authenticated as a legitimate reader–server unit. In the authentication process, A needs to send Auth_1_ = Hash(N||R_i_) and Auth_3_ = Hash(N’||R_i_) to the tag for verification. To generate Auth_1_ and Auth_3_ with non-negligible probability, A needs to know the secret parameters N and R_i_, which are unknown to A by Lemma 1. Hence, A can impersonate as a legitimate reader–server unit with negligible probability.To sum up, the adversary A can neither impersonate as a valid tag nor a valid reader–server unit with non-negligible probability. Thus, the protocol is shown to provide secure mutual authentication.
**Q.E.D.**


#### 5.2.3. Desynchronization Attacks

**Theorem** **4.**
*The proposed protocol can defend against desynchronization attacks.*


**Proof:** In the proposed protocol, server S updates the session identity SID_i_^T^ to SID_i+1_^T^ to after receiving the message M_4_. If the messages M_4_ is blocked, server S and tag T can still synchronize with each other using SID_i_^T^. Tag T will update SID_i_^T^ to SID_i+1_^T^ once the message M_5_ is received and verified. Thus, we assume that the adversary A calls the Block oracle to block the message M_5_ so that tag T will not update the session identity while server S does. To deal with this issue, server S still stores the old entry {SID_i_^T^, C_i_, R_i_} for tag T in the database. Once server S launches a new authentication phase for tag T, tag T will respond with {SID_i_^T^} so that the authentication can continue. In this way, the protocol can ensure the resilience of desynchronization attacks.
**Q.E.D.**


#### 5.2.4. Physical Attacks

**Theorem** **5.**
*The proposed protocol is safe against physical attacks.*


**Proof:** The physical attack can be modeled as the adversary A invokes the Reveal oracle to obtain all the data from the tag memory. Unlike most of the existing RFID authentication protocols, in the protocol, the tag stores no secret security credentials (i.e., keys) in its memory. Even with the Reveal oracle, A cannot access the secret parameters. Therefore, the proposed protocol can be regarded safe against physical attacks.
**Q.E.D.**


#### 5.2.5. Clone Attacks

**Theorem** **6.**
*In the proposed protocol, tags are unclonable.*


**Proof:** PUF techniques can prevent the clone attack since a PUF circuit cannot be forged [35]. As mentioned in Section 4.2, in the proposed protocol, each tag is embedded with a PUF circuit. Hence, we can claim that tags in the protocol are unclonable.
**Q.E.D.**


#### 5.2.6. Modeling Attacks

**Theorem** **7.**
*The proposed protocol is immune to modeling attacks of PUF.*


**Proof:** The modeling attack of a PUF is based on enough challenge–response pairs of the PUF [50]. In the proposed protocol, the adversary A can obtain the challenge C_i_ from message M_3_ while its response R_i_ is encrypted to N’ by a random number N. Thus, A cannot directly collect CRPs. The adversary A is able to gather enough (C_i_, N’)-pairs by eavesdropping. However, N’ is generated based on only the challenge C_i_ but also the random number N which is different in each round of the authentication. Hence, the adversary A cannot correctly derive a numerical model due to the randomness introduced by N.
**Q.E.D.**


### 5.3. Formal Security Verification Using Scyther Tool

We evaluate our proposed protocol based on ideal PUF by a formal verification tool, Scyther [51], which is a widely accepted tool for the automatic verification of security protocols. For a target protocol, it is able to provide the analysis result of complex attack scenarios efficiently. Hence, we utilize Scyther to further verify the security correctness of the proposed protocol.

According to the proposed protocol, we define two roles, reader–server unit S and tag T, in Scyther. Then we implement each role with its corresponding behavior. In the proposed protocol, there exists four security parameters: N, M, R_i_, and R_i+1_, which should not be exposed. Thus, we make *Secret* claims on them. In the Scyther tool, we set the search pruning to “Find all attacks” and the matching type to “Find all type flaws”. The outcome of the formal security verification of the proposed protocol is summarized in Table 2 which shows that the proposed protocol is safe. The details of the implementation of our proposed protocol is presented in Appendix A.

## 6. Performance Analysis

In this section, we analyze the performance of our proposed protocols by comparing with some of the recently proposed PUF-based RFID authentication protocols [40,41,42,44,45,46,47] discussed in Section 2.

### 6.1. Security Performance

Firstly, we compare the security performance of our proposed protocols with them based on some common vulnerabilities in RFID systems, as shown in Table 3. From Table 3, we can see that all the protocols have at least one vulnerability except the protocols [42,47] and our proposed protocols. For example, since most of the protocols require to store secrets in tag memory, they are vulnerable to physical attacks. An attacker can easily trace back previous communications once the secrets are obtained. Thus, such protocols that are vulnerable to physical attacks cannot guarantee forward secrecy either. 

### 6.2. Efficiency Performance

In addition to security, the efficiency performance should also be taken into consideration by RFID authentication protocols. Since RFID tags usually have limited resources while backend servers have enough resources, we focus on the computation and storage on tags, the bandwidth during authentication and the scalability when comparing the efficiency performance. We assume the security level to be achieved is 128-bit. The comparison results are presented in Table 4.

In our ideal PUF-based protocol, the tag performs most of the operations at Step 4. At Step 4, the tag needs to invoke the hash function three times for the verification of Auth_1_ and for the computation of C_i+1_ and Auth_2_. Besides, the PUF is used twice to generate R_i_ and R_i+1_. The tag also uses its RNG to produce a random number M. The rest operations, two calling of the hash function, are done at Step 6 to verify Auth_3_ and to compute SID_i+1_^T^.Thus, the computation operations for the tag in our ideal PUF-based protocol are hash function five times, PUF twice and RNG once while our noisy PUF-based protocol needs one additional fuzzy extractor reconstruction at Step 4 and one more hash operation at Step 6. From Table 4, we can see that the computation overhead of our proposed protocols is similar to existing protocols in the same environment.

For the storage overhead comparison, the tag in our ideal PUF-based protocol just needs to store the session identity. Thus, 128 bits of storage cost is needed. Our noisy PUF-based protocol requires the tag to store the session identity and the help data. According to Asyu et al. [41], for a 128-bit security level, the length of the help data is 1264-bit. So the storage overhead is 128 + 1264 = 1392 bits. As demonstrated in Table 4, our proposed protocols bear lowest storage overhead among all protocols in the same environment.

When comparing the bandwidth, we omit the auxiliary fields in a packet and just consider the length of data. In our ideal PUF-based protocol, the messages M_3_ and M_4_ have the same maximum data length, which is 384 bits. Therefore, the bandwidth of our protocol is similar to that of other protocols. In our noisy PUF-based protocol, the longest message is M_5_, whose length is 1392 bits. Thus, our protocol requires smaller bandwidth than Asyu et al. [41] and Huth el al. [42], while the bandwidth of Gope et al. [47] is the same as that of our protocol.

Furthermore, we also consider the scalability of our protocols and other protocols. As discussed in Section 2, the protocols [40,41,42] have to perform exhaustive search in the backend server to identify a tag. Thus, these protocols are not applicable in case of large scale database with numerous tags while our protocols do not need exhaustive search and are suitable for large scale IoT applications hence.

### 6.3. Highlights of Our Protocols

In the ideal PUF environment, Table 3 shows that only the protocol of Gope et al. [47] and ours can satisfy all the security requirements. When comparing with their protocol, our ideal PUF-based protocol suffers more overhead in terms of computation (one more hash operation) and bandwidth (64 bits more) as demonstrated in Table 4. However, the protocol of Gope et al. requires the tag to store several (i.e., n) unlinkable 64-bit pseudo-identities for the goal of defending against desynchronization attacks, which results in considerably more storage overhead than our protocol. Furthermore, since each desynchronization will cost one pseudo-identity, at most n – 1 pseudo-identities can be used. After that, the tag should authenticate itself using the last pseudo-identity and reload a new set of pseudo-identities from the backend server. If the reloading fails (i.e., interrupted by an attacker) and the pseudo-identities are used up, the tag has to be registered into the backend server again through a secure channel. All these operations incur computation and communication overhead while our protocol will not introduce any extra overhead due to the desynchronization. In addition, the protocol of Gope et al. requires the tag to send a message first in the authentication phase. Such behavior is only available for active tags that are expensive and less used [52]. In our protocol, the reader sends a query first when an authentication launches, which is suitable for low-cost tags such as passive tags and semi-passive tags.

In the noisy PUF environment Table 3 illustrates that; besides, the protocol of Gope et al. and ours, the protocol of Huth et al. [42] can also meets all the security properties. The protocol of Huth et al. needs to perform a CBKA phase before the actual authentication phase, which results in additional overhead. In addition, the protocol relies on exhaustive search to identity a tag so that it cannot support scalability property. On the other hand, by comparing our proposed protocol with the protocol of Gope et al., we can see that the only advantage of the protocol of Gope et al. is that the protocol needs one less hash operation in terms of the computation overhead whereas the protocol incurs even more storage overhead than its ideal PUF-based version: in the ideal PUF case, the protocol of Gope et al. needs 50n% more storage than ours while, in the noisy PUF case, it requires 100n% more storage. Moreover, similar to the ideal PUF case, the protocol needs to perform extra operations when facing the desynchronization while ours do not. Furthermore, the protocol is only available for actives tags as its ideal PUF version while ours can support low-cost tags.

In summary, comparing with existing protocols, our proposed protocols can guarantee all the security requirements with moderate computation cost and communication cost. Since our protocols do not require any secret parameter to be stored on the tag, our protocols incur the lowest storage cost in the same environment. Due to not using exhaustive search for the validation of tags, our protocols ensure the scalability property. In addition, our protocols can also work on low-cost tags. 

## 7. Implementation Discussion

In the section, we discuss the implementation cost for our proposed protocols for a security level of 128-bit. Consider that a backend server has much more computation and storage resources than a RFID tag, we focus on the implementation of the tag. 

### 7.1. The Proposed Protocol Based on Ideal PUF

In our ideal PUF- based proposed protocol, the security primitives involved include a RNG, a hash function, and an ideal PUF. Since a EPC Gen2 tag should be equipped with a RNG according to Section 6.3.2.7 in EPC Gen2 specification [3], we just need to implement a hash function and an ideal PUF. Due to the limited resource on a tag, a lightweight hash function should be adopted. For instance, as mentioned in Section 1, it is possible to implement a 128-bit SPONGENT with 1060 logical gates. There exist lots of PUF implementations, such as arbiter PUF, ring oscillator PUF, SRAM PUF, and many other variations. Unfortunately, most of them are noisy PUFs requiring error correction mechanisms. In 2017, Wang et al. [53] proposed Locally Enhanced Defectivity Physical Unclonable Function (LEDPUF), which is a stability-guaranteed PUF that needs no error correction techniques. In Wang et al. [53], weak LEDPUF and strong LEDPUF were both presented. As discussed in Section 5.2.6, our proposed protocol can defend against modeling attacks. Thus, weak LEDPUF is a better choice for our protocol in regard of the implementation cost. According to Section 3 in Wang et al. [53], the weak LEDPUF requires 3 transistors to generate a bit response. Thus, for a 128-bit PUF response, 384 transistors are required to implement the LEDPUF. 

### 7.2. The Proposed Protocol Based on Noisy PUF

In our noisy PUF-based proposed protocol, the security primitives involved include a RNG, a hash function, a noisy PUF, and a fuzzy extractor. Considering the various implementations of noisy PUFs and fuzzy extractors, we can take the implementation of Aysu et al. [41] as a reference. In their protocol, the tag should equip with an SRAM PUF, an SRAM TRNG, a pseudorandom function (PRF) based on the SIMON block cipher and a fuzzy extractor constructed by a BCH error corrector and a PRF based strong extractor. Aysu et al. listed the hardware utilization of their implementation, which requires 3543 lookup tables, 1275 registers, and eight blocks of RAM on a Xilinx Virtex-5 FPGA (XC5VLX30) to offer a security level of 128 bits. They also stated that their protocol can fit into a small microcontroller.

With implementation of Aysu et al. at hand, the SRAM PUF, the TRNG, and the fuzzy extractor can be reused in our protocol. Thus, we only need to consider the implementation cost of an additional hash function. According to Van Herrewege et al. [54], a 128-bit SPONGENT hash function can be implemented with 160 lookup tables and 146 registers on the same hardware platform as the implementation of Aysu et al. Therefore, we can roughly estimate the implementation cost of our protocol, which includes 3703 lookup tables, 1421 registers, and eight blocks of RAM on a Xilinx Virtex-5 FPGA (XC5VLX30).

## 8. Conclusions

In this paper, we first proposed a lightweight RFID mutual authentication protocol for ideal PUF. Then we presented an improved protocol, which can support noisy PUFs. We analyzed the security and the performance of our proposed protocols. The analysis results show that our protocols can ensure the desired security properties efficiently. Especially, our protocols are immune to physical attacks and clone attacks since the tag stores no secrets and equips with a PUF. Besides, our protocols support low-cost tags and the applications with a large amount of tags. Therefore, our proposed protocols are more suitable to enhance the security and privacy for IoT applications.

## Figures and Tables

**Figure 1 sensors-19-02957-f001:**
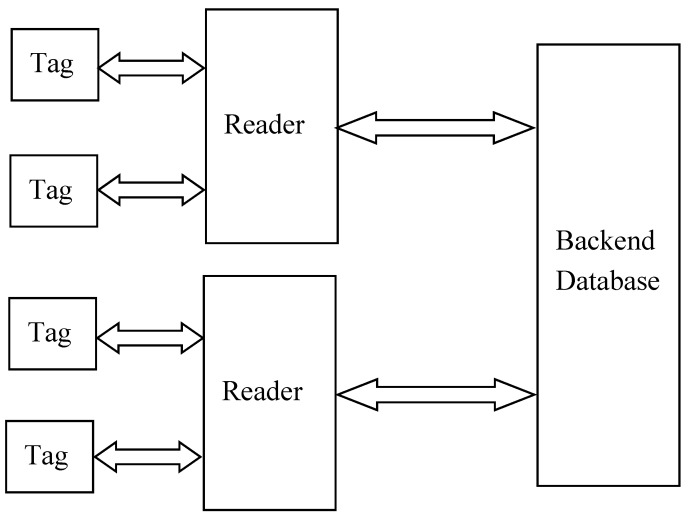
A typical radio frequency identification (RFID) system.

**Figure 2 sensors-19-02957-f002:**
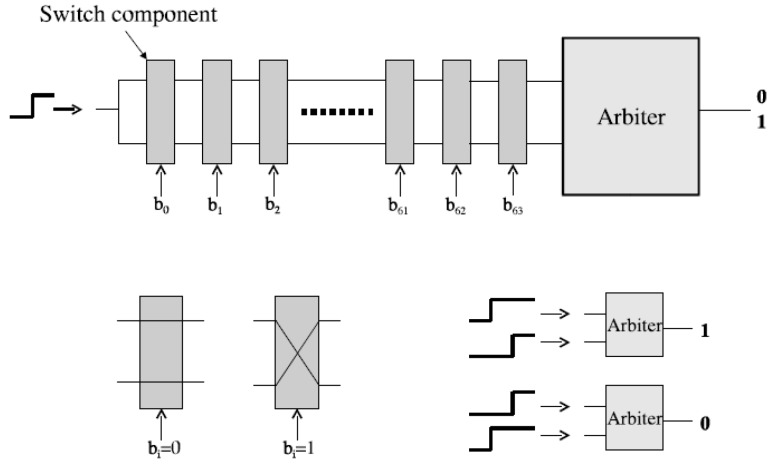
Arbiter physically unclonable function (PUF) circuit diagram.

**Figure 3 sensors-19-02957-f003:**
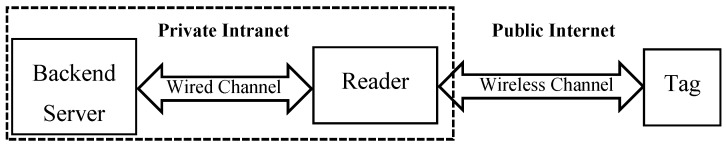
Communication channels in a typical RFID system.

**Figure 4 sensors-19-02957-f004:**
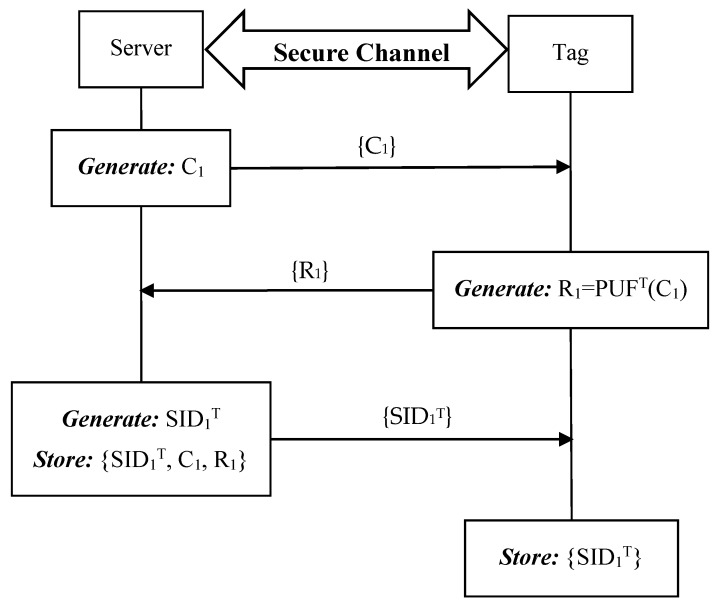
Setup phase of the proposed ideal PUF-based protocol.

**Figure 5 sensors-19-02957-f005:**
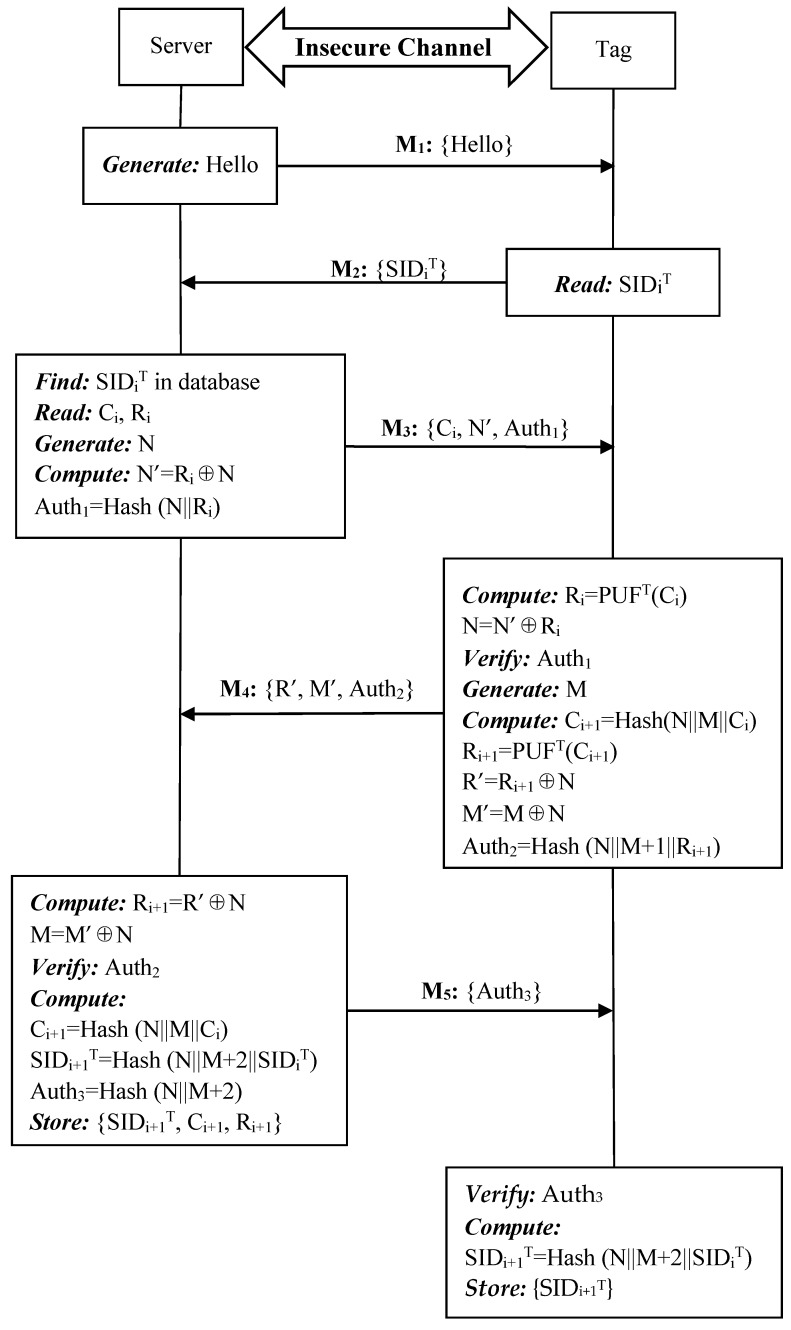
Authentication phase of the proposed ideal PUF-based protocol.

**Figure 6 sensors-19-02957-f006:**
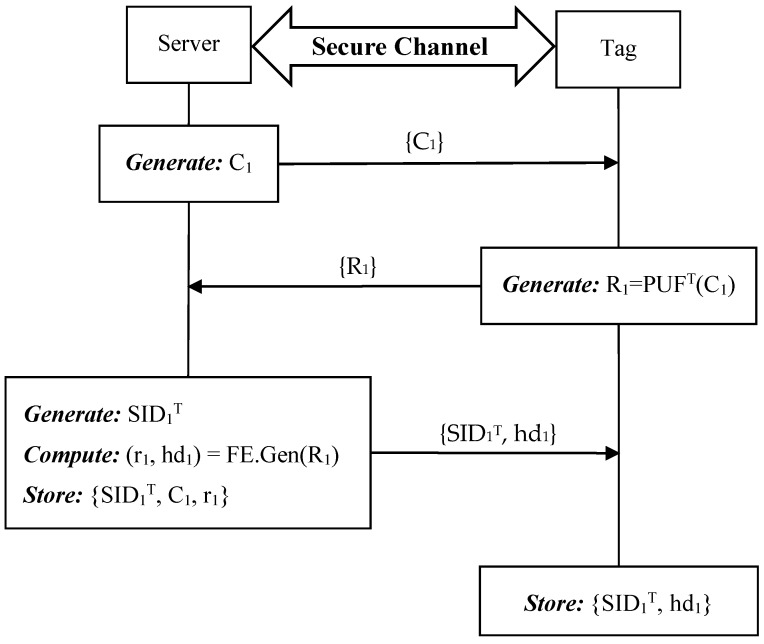
Setup phase of the proposed noisy PUF-based protocol.

**Figure 7 sensors-19-02957-f007:**
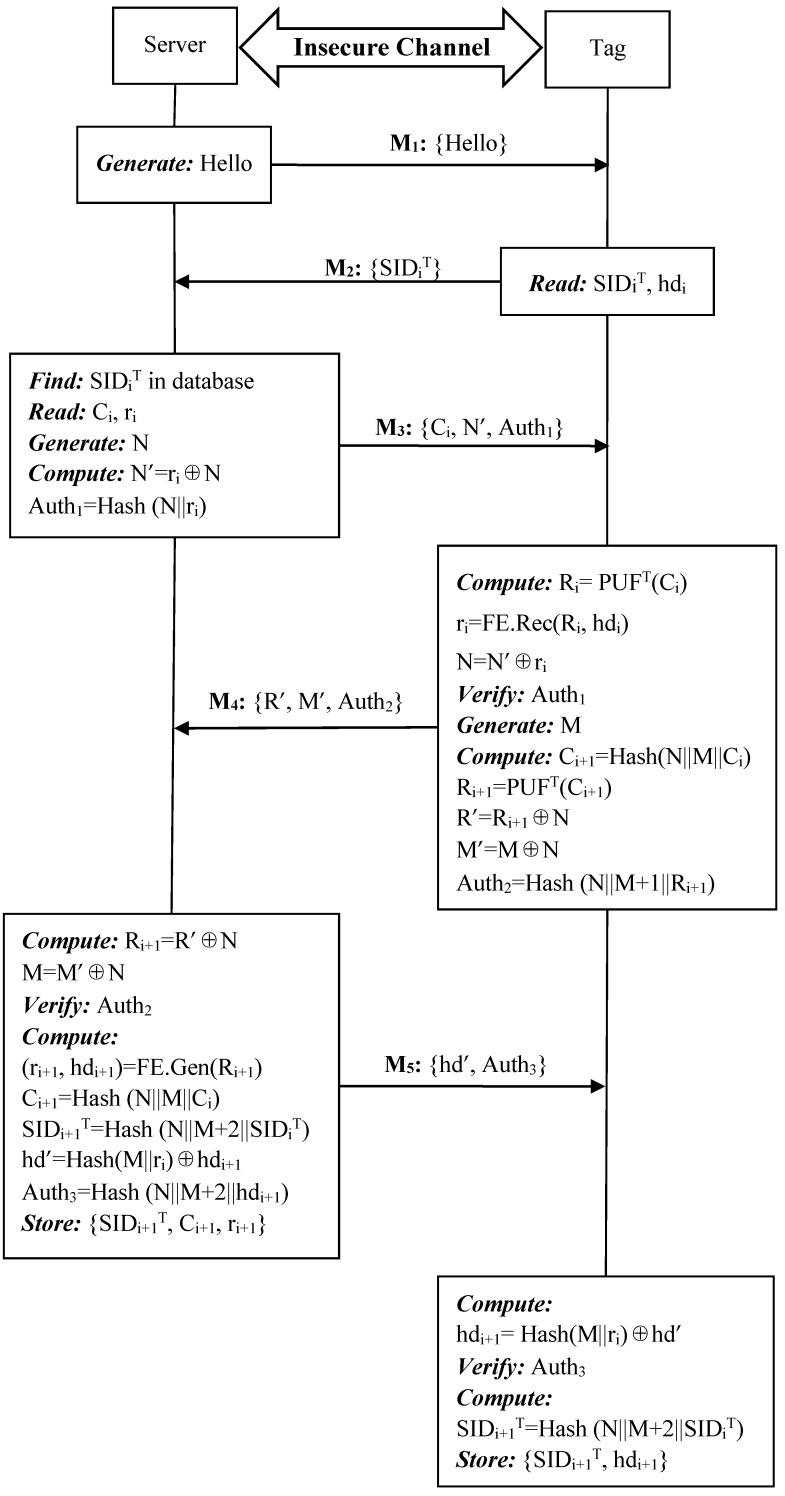
Authentication phase of the proposed noisy PUF-based protocol.

**Table 1 sensors-19-02957-t001:** Operations and cryptographic functions.

Symbol	Definition
⊕	Exclusive OR operation
||	Concatenation operation
PUF^T^	The physically unclonable function on tag T
Hash	One-way hash function shared by tag T and server S

**Table 2 sensors-19-02957-t002:** The verification outcome of our proposed protocol using Scyther.

Claim				Status	Comments
ideal_PUF	S	ideal_PUF,S1	Secret N	OK	No attacks within bounds
		ideal_PUF,S2	Secret R_i_	OK	No attacks within bounds
		ideal_PUF,S3	Niagree	OK	No attacks within bounds
		ideal_PUF,S4	Nisynch	OK	No attacks within bounds
		ideal_PUF,S5	Alive	OK	No attacks within bounds
		ideal_PUF,S6	Weakagree	OK	No attacks within bounds
	T	ideal_PUF,T1	Secret M	OK	No attacks within bounds
		ideal_PUF,T2	Secret R_i+1_	OK	No attacks within bounds
		ideal_PUF,T3	Niagree	OK	No attacks within bounds
		ideal_PUF,T4	Nisynch	OK	No attacks within bounds
		ideal_PUF,T5	Alive	OK	No attacks within bounds
		ideal_PUF,T6	Weakagree	OK	No attacks within bounds

**Table 3 sensors-19-02957-t003:** Security performance comparison.

Vulnerability	Ideal PUF-based Protocols						Noisy PUF-based Protocols			
	[40]	[44]	[45]	[46]	[47]	Ours	[41]	[42]	[47]	Ours
Traceability							√			
Lack of Forward Secrecy	√	√	√	√						
Impersonation										
Desynchronization		√	√							
Physical Attacks	√	√	√	√						

**Table 4 sensors-19-02957-t004:** Efficiency performance comparison.

	Protocols	Computation	Storage (bits)	Bandwidth (bits)	Scalability
**Ideal** **PUF-Based** **Protocols**	[40]	4Hash + 2PUF + RNG	512	384	No
	[44]	3LFSR + 2PUF	512	384	Yes
	[45]	2PUF + RNG	384	256	Yes
	[46]	5Hash + 6PUF + RNG	512	384	Yes
	[47]	4Hash + 2PUF + RNG	128 + 64n *	320	Yes
	Ours	5Hash + 2PUF + RNG	128	384	Yes
**Noisy** **PUF-Based** **Protocols**	[41]	3Hash + 2PUF + RNG + SKE + FE.Gen	192	2168	No
	[42]	3Hash + 2PUF + RNG + SKE + CBKA + FE.Rec	1804	2168	No
	[47]	5Hash + 2PUF + RNG + FE.Rec	1456 + 1392n *	1392	Yes
	Ours	6Hash + 2PUF + RNG + FE.Rec	1392	1392	Yes

Hash: Hash Function; PUF: Physical Unclonable Function; RNG: Random Number Generator; LFSR: Linear Feedback Shift Register; SKE: Symmetric Key Encryption; CBKA: Channel-based Key Agreement Operations; FE.Gen: Fuzzy Extractor Generation; FE.Rec: Fuzzy Extractor Reconstruction. * For the protocols in Gope et al. [47], n is the number of pseudo-identities stored in the tag.

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
