# Peer review of "A Lightweight RFID Mutual Authentication Protocol with PUF"

_sensors, 2019, doi:10.3390/s19132957_

Round 1
Reviewer 1 Report
The paper was intereting but with using checkmarks instead it make one wonder if its a viable project. I would also think about reliability it was missing.
Author Response
Point 1: The paper was interesting but with using checkmarks instead it makes one wonder if it’s a viable project. I would also think about reliability it was missing.
Response 1: We evaluate our proposed protocol based on ideal PUF by a formal verification tool, Scyther. The outcome is presented in Section 5.3. Considering the reliability of PUF, we propose an enhanced protocol for noisy PUFs in Section 4.5. The implementations of our protocols are discussed in Section 7.

Reviewer 2 Report
This paper presents an interesting privacy-preserving RFID authentication protocol based on PUF and compares it with prior works.
How resistant is this design against modeling attacks of PUF? Please include a security analysis.
How will you deal with noisy PUF outputs?
Author Response
This paper presents an interesting privacy-preserving RFID authentication protocol based on PUF and compares it with prior works.
Point 1: How resistant is this design against modeling attacks of PUF? Please include a security analysis.
Response 1: Our proposed protocol can defend against the modelling attacks. The security analysis presented in Section 5.2.6.
Point 2: How will you deal with noisy PUF outputs?
Response 2: We proposed an enhanced protocol based on noisy PUF in Section 4.5, which leverages a fuzzy extractor to handle the noisy PUF outputs.

Reviewer 3 Report
The article introduces a lightweight RFID authentication protocol that is based on a few basic and known solutions: PUF, CRC and RNG. Furthermore, performance, security and privacy analyses of the protocol follow.
The advantages of the paper are that the protocol is simple therefore lightweight, the description and the explanation are clear enough to understand the idea, the analysis of various scenarios looks sound and I appreciate the attempt to make an extensive comparison.
On the other hand the major weaknesses concern poor language and lack of thorough analyses --- especially concerning the tools that the authors propose to use. Here are my remarks:
1) The authors base the security on PUF, however most PUFs are not so perfect (not very stable, not very reliable etc.) --- i.e. the solution presented as an example. The problem with PUF implementation is that the responses are not certain and here --- if one wraps that in CRC code and performs an authentication --- many of them will fail if one asseses the result binary. Usually there is a need of correction of the PUF responses or an evaluation how far is a response from the expected one.
2) The authors use a few times RNG. What are the expectations concerning RNG?
TRNG? PRNG? How good one? What kind of vulnerabilities introduces a weak RNG?
3) The authors mentioned ”physical attack”. I’m guessing based on [5] that as a side-channel attack. However physical attacks are just a part of many SCAs and the physical ones are usually the most expensive and difficult to perform in practice. What about other SCAs?
4) What happens in case when Auth1 in step M4 fails? The tag does not compute the rest? Does it even answer?
If yes, there is possibility for the attacker to feed many Ci-Ri pairs to the tag (generating N and computing CRC) and checking one by one if a response fits to the challenge (brute force checking).
If no, the same scenario, but with timing attack, which was not consider here.
5) Have the authors considered complex attacks (for example repetition attack on CRC in M3 after blocking M5)?
6) Why do you assume the wire connection (line: 164)? Did you mean secure? Wire is not much more secure then air.
7) I think that the assumption for the server of “no limitation on computation or storage” may be a little overkill...
8) I’m glad that the authors included comparisons to other protocols, however I think they should be much more detailed.
9) The language must be significantly improved (preferably by a native speaker who understand the subject). Just a few examples of unfortunates:
- one of the key technique
- logical gates available can be used for the security
- In 2002, the first PUF is invented.
- PUF techniques can has good resistance
- cryptograph function
- with the transmitted nonce until find a match
10) The style must be improved - a few unfortunates:
- the adversary “A” is written different on page 7 and half of 8 and the rest of the article
- many variables are sometimes written in italics, sometimes not
- “Internet of things (IOT)” - line 85
In conclusion, I find the article interesting and having some journal potential, however I would consider it more as a conference paper.
Reviewer 4 Report
The manuscript proposed a PUF-based authentication protocol for RFID devices. Major concerns can be found below:
1. The paper needs further proof read to correct grammar mistakes and inappropriate terms and phrases. For example, the sentence “EPC Gen2 tags is the mainstream for the develop RFID applications” needs to be revised at several places.
2. In sections 4.2 the assumption 2 is not appropriate since the reliability of regenerating the same responses for PUFs depends on the strength and overhead of the error correction mechanism. The authors need to provide more performance and overhead information of the error correction scheme before making the assumption.
3. In section 5.2.1 Theorem 2, it is incorrect to state that “A cannot inverse the CRC function” because unlike hash, CRC can be easily reversed. Therefore, the security of reverse-engineering the inputs (which are the secrets) of the CRC function from its plain outputs in the whole protocol needs to be re-evaluated, so are the validity, performance and overhead of the overall proposed protocol.
4. Step 3 states that “Upon receiving the session identity SIDiT from T, S uses it as an index to search a matched entry.”, which indicates that the proposed protocol still needs exhaustive search at the server. It is contradictory to what the authors claimed in section 6 that the protocol does not need exhaustive search and is thus scalable.
5. The paper lacks experimental implementations to evaluate and validate the cost, performance and the scalability of the proposed protocol, as can be referred to in reference [42]. The authors need to add this part as the experimental evaluation part.
6. In section 4.2, it is incorrect to state that “A PUF can be considered as a challenge-response pair (CRP).” since a PUF instance typically maps a set of challenges to a set of responses.
Author Response
Point 1: The paper needs further proof read to correct grammar mistakes and inappropriate terms and phrases. For example, the sentence “EPC Gen2 tags is the mainstream for the develop RFID applications” needs to be revised at several places.
Response 1: We have proof read our manuscript.
Point 2: In sections 4.2 the assumption 2 is not appropriate since the reliability of regenerating the same responses for PUFs depends on the strength and overhead of the error correction mechanism. The authors need to provide more performance and overhead information of the error correction scheme before making the assumption.
Response 2: We revise our manuscript as follows. We first propose a lightweight RFID mutual authentication protocol based on ideal PUFs in Section 4.4. Then we enhance this protocol so that it can be adopted with noisy PUFs in Section 4.5.
Point 3: In section 5.2.1 Theorem 2, it is incorrect to state that “A cannot inverse the CRC function” because unlike hash, CRC can be easily reversed. Therefore, the security of reverse-engineering the inputs (which are the secrets) of the CRC function from its plain outputs in the whole protocol needs to be re-evaluated, so are the validity, performance and overhead of the overall proposed protocol.
Response 3: We now use a hash function instead of the CRC function. The relevant parts are also revised.
Point 4: Step 3 states that “Upon receiving the session identity SIDiT from T, S uses it as an index to search a matched entry.”, which indicates that the proposed protocol still needs exhaustive search at the server. It is contradictory to what the authors claimed in section 6 that the protocol does not need exhaustive search and is thus scalable.
Response 4: Here the exhaustive search indicates that the backend server has to perform some operations (i.e., hash function or cryptographic function) before checking every record in the database. As discussed in Section 3, according to [41], if the server can find the identity of tag by directly checking the received data, the time cost can be constant. In our protocols, the server can locate the tag in the database by directly checking received SID.
Point 5: The paper lacks experimental implementations to evaluate and validate the cost, performance and the scalability of the proposed protocol, as can be referred to in reference [40]. The authors need to add this part as the experimental evaluation part.
Response 5: In Section 5.3, we use a formal verification tool to validate the security correctness of our proposed protocol. In Section 7, the implementation of our proposed protocol is discussed.
Point 6: In section 4.2, it is incorrect to state that “A PUF can be considered as a challenge-response pair (CRP).” since a PUF instance typically maps a set of challenges to a set of responses.
Response 6: We revised the description. In addition, a noisy-PUF based protocol is proposed in Section 4.5.

Round 2
Reviewer 3 Report
I’m glad to see that the article has significantly improved, due to the use of hash function and by taking into consideration the noisy PUFs (as well as the language).
There is something unclear in the new sentence:
“For example, an arbiter PUF requires roughly 8 logical gates per input bit, plus 4 logical gates for the arbiter. Thus, a 128-bit arbiter PUF needs about 1,028 logical gates (...)”.
Such an arbiter would have just one bit of the response, unless you assume that there is some range of input sequence used in order to generate a sequence of response bits.
I did not get it how do you use such a 1,028 logical gates big structure in the challenge-response manner. What is the size of a PUF-challenge here and what is the size of a PUF-response?
Author Response
Such a circuit will just generate one bit of the response. According to Herder et al. [36, Section IV.C], to construct a k-bit response, a linear feedback shift register (LFSR) is used to generate a pseudorandom sequence based on the input challenge. The PUF is then evaluated k times using k different bit vectors from this larger pseudorandom sequence. In the example, the challenge size is 64-bit while the response size is not fixed. We revised the sentence you mentioned and added related discussion to Section 2.
Reviewer 4 Report
1. In the authors’ response to original comment 3, the CRC module is replaced with the hash function. Even though corresponding comments and reference are added that there exist some lightweight hash functions, it is very unclear what the unique contributions are of the proposed protocol over existing related protocols which also only requires the hash module, for example, compared to reference: P. Gope, J. Lee, and T~Q. S. Quek, “Lightweight and Practical Anonymous Authentication Protocol for RFID Systems Using Physically Unclonable Functions,” IEEE Transactions on Information Forensics and Security, vol. 13(11), pp. 2831-2843, 2018. The authors need to add a separate section that clearly states the unique contributions and advantages of the proposed protocol over other related ones.
2. The biggest concern has not been appropriately addressed at all in the revised version, i.e., the implementation evaluation of the proposed protocol based on “Noisy PUF” is missing. The authors arbitrarily stated that “the implementation cost of our protocol can be A LITTLE BIT LOWER THAN [39]” without conducting any hardware validations at all. According to what is reported for the “ideal PUF” case, the proposed protocol has even larger overhead compared to the work: P. Gope, J. Lee, and T~Q. S. Quek, “Lightweight and Practical Anonymous Authentication Protocol for RFID Systems Using Physically Unclonable Functions,” IEEE Transactions on Information Forensics and Security, vol. 13(11), pp. 2831-2843, 2018. Detailed comparisons regarding overhead as well as unique contributions of the proposed protocol with the abovementioned work and other works need to be provided.
